# An unexpected confounder: how brain shape can be used to classify MRI scans ?

Valentine Wargnier-Dauchelle      VALENTINE.WARGNIER@CREATIS.INSA-LYON.FR
Thomas Grenier            THOMAS.GRENIER@CREATIS.INSA-LYON.FR
Michaël Sdika             MICHAEL.SDIKA@CREATIS.INSA-LYON.FR
*INSA Lyon, Universite Claude Bernard Lyon 1, CNRS, Inserm, CREATIS UMR 5220, U1294, Lyon, France*

**Editors:** Accepted for publication at MIDL 2024

## Abstract

Although deep learning has proved its effectiveness in the analysis of medical images, its great ability to extract complex features makes it susceptible to base its decision on spurious confounders present in the images. However, especially for medical applications, network decisions must be based on relevant elements. Numerous confounding factors have been identified in the case of brain scans such as gender, age, MRI sites or scanners, etc. Nevertheless, although skull stripping is a classic preprocessing step for brain scans, brain shape has never been considered as a possible confounder. In this work, we show that brain shape is used in the classification of brain MRI scans from different databases, even when it should not be considered as a clinically relevant factor. To this purpose, we introduce a rigorous two steps method to assess whether a factor is a confounder or not, and we apply it to identify the brain shape as a confounding variable in brain images classification. Lastly, we propose to use a deformable registration in the data preprocessing pipeline to align the brain contours of the images in the datasets, whereas standard pipelines often do nothing more than affine registration. Including this deformable registration step makes the classification free from the brain shape confounding effect.

**Keywords:** Confounding factor, Classification, Brain shape, Deformable registration, Interpretability

## 1. Introduction

Deep learning has emerged as a powerful tool in the field of medical imaging. Its ability to automatically learn and extract complex patterns from vast amounts of data has revolutionized the way we analyze images. However, the great performances of deep learning come with the price of the black-box nature of these methods: deep neural networks, with their non-linearity and their large number of parameters, are difficult to explain. Training explainable and interpretable networks is therefore a key issue for medical image analysis as the lack of transparency can hide the fact that the network decision may be based on wrong reasons: a bias in the training set can make a confounding factor plays an important role in the decision. Classifiers are especially subject to this problem whether they are used for pure classification problems or as guidance for adversarial networks or diffusion models.

Several tricks can be used to reduce or remove the influence of a known confounding factor from a model. The simplest is to carefully collect the training dataset such that the confounding variables are matched in the different classes as in (Leming et al., 2022). However, this approach makes it tedious to create large datasets. If possible, normalization

preprocessing can also be used to discard variations of this variable across the dataset. For example, in (Wargnier-Dauchelle et al., 2021), the MRI signature is removed using brain tissue probability maps instead of MRI scans as input of a deep classifier. Data augmentation can also be used to make the model invariant to a set of variables or transforms. In the literature, works have also been proposed to train models free from the influence of a known confounder. In (Zhao et al., 2020), the features of a network are trained for the prediction of the objective task but also trained adversarially for the prediction of the gender, considered here as the confounding variable, making the features invariant to this confounder. In (Wang et al., 2018), the model is first trained for the objective task and the top layer is then fine-tuned to predict the confounding variable: the gender, the subject or the contrast material. During this fine-tuning step, weights sensitive to the confounding factor are identified and discarded. For these methods to be used, the potential confounders need to be identified first. Moreover, for the two latter methods, the potential confounding factor should be available as a scalar of categorical data and the value of the confounding variable should be known during training. To identify confounding factors in images, attribution maps can be used to localize the information used by the network to make its decision. For example, in (Sun et al., 2023), an attribution maps comparison protocol was proposed by visualizing confounding factors artificially added to the images. In (Wargnier-Dauchelle et al., 2021, 2023), these maps are used to validate and/or improve the interpretability of the network by verifying that the decision is based on brain lesions.

In this work, we assess the importance of the brain shape as a confounding factor in the classification of brain images. To do so, we propose a rigorous protocol to verify that the brain shape is indeed a confounding factor used as a part of the network decision despite the standard affine spatial normalization. The first step of our protocol is to verify if it is possible to classify various datasets using the confounding factor only: for the concrete case we investigate, we propose to use the brain mask as a brain shape representation and our first non-intuitive result is that it is possible to classify the datasets using only this mask. Then, we check that the brain shape is indeed part of the decision when original grayscale images are used as input of the network. To this end, we modify the image such that the identified factor could be from the opposite class and evaluate the impact on the classification task. We come up with two solutions to change the brain shape from one class to another: by trimming the borders of the brain or with deformable registration. Finally, we show that by complementing the standard affine registration of the preprocessing with a deformable registration to normalize the brain shape, we can classify brain images while canceling out the confounding effect of brain shape in classification.

## 2. Method

### 2.1. A generic two steps confounding factor identification procedure

Our procedure to assess that a variable is indeed a confounding factor is in two steps. In the first step, we verify that the suspicious variable is a potential confounding factor by checking whether or not it is possible to classify the data using this variable only. To do so, a classifier, having this variable only as input, is trained to classify the subjects using the same class label as the original problem. If the classifier is random, with an accuracy close to 0.5, the variable can be discarded from the potential confounders of the problem.

If the classification is possible in the first step, we check in the second step whether this factor is indeed used by a model trained with the original images as input. To do so, the model is trained conventionally on the original images. Then, test data are transformed such that the value of the suspicious factor lies within the distribution of this factor for the opposite class, while modifying the image as little as possible. The difference in classification performances between the original and the transformed test data is then measured. A lower classification performance for transformed data would indicate that the identified factor is used by the model to correctly classify and is indeed a confounder.

### 2.2. Identifying the brain shape as a confounding factor

For the concrete case of the brain shape, the first step is achieved by trying to classify several brain datasets using the brain masks only as input of the network. This binary mask, indicating whether a voxel is inside the brain (value of 1) or outside (value of 0), is used as a representation of the brain shape. For the second step, we need a transform to make the brain shape of a subject match the shape of subjects in the opposite class, while changing the image content as little as possible. Two such transforms are investigated: brain mask crop or brain mask registration.

**Brain mask crop**   A brain mask is randomly drawn from the opposite class and used to crop the grayscale image of the current subject: pixels outside the mask are set to the background value. This cropped image is given as input to the classification model at test time. The image remains the same inside the brain mask but it is changed at its border. Note that this technique modifies only a part of the shape: the part outside the mask drawn from the opposite class.

**Brain mask registration**   A brain mask is also randomly drawn from the opposite class but this time, it is used as a reference to realign the brain shape of the current subject. To realign a moving brain mask $B_m$ on a reference brain mask $B_r$, we solve the following optimization problem:

$$\min_{T \text{ s.t. } J(x) \geq t} \sum_{x \in \partial B_r} d(T(x), \partial B_m) + \lambda \sum_x ||\Delta T(x)||^2, \qquad (1)$$

where $T$ is the transformation we look for, $\partial B_m$ and $\partial B_r$ are the border of the two brain masks and $d(., \partial B_m)$ is the Euclidean distance to the border of the moving brain mask. To penalize strong deformations, a bending energy term with a coefficient $\lambda$ is added and the Jacobian of the transformation $J(x)$ is constrained to be higher than a given threshold $t$ for all voxels. Any registration algorithm could be used with a null image as the fixed image, the distance transform of $\partial B_m$ as the moving image and a cost function masked with $\partial B_r$. The transformation $T$ is then applied to the original grayscale image to obtain an image with a brain shape of the opposite class.

### 2.3. Eliminating the brain shape confounding effect with normalization

Affine registration to a reference template is usually included in the data preprocessing to spatially normalize the datasets. We advocate that this affine registration step is not sufficient to avoid the brain shape confounding effect in deep learning. We propose to add

Table 1: T1 MRI datasets. H refers to healthy, MS to multiple sclerosis, T to tumors and AD to Alzheimer's Disease

| Dataset | IXI | HCP | MPI | kirby | IBC | OFSEP | BraTS | ADNI | |
|---------|-----|-----|-----|-------|-----|-------|-------|------|------|
| | | | | | | | | CN | AD |
| $N_{train}$ | 400 | 500 | 64 | 22 | 8 | 383 | 280 | 183 | 150 |
| $N_{val}$ | 130 | 100 | 15 | 5 | 2 | 97 | 40 | 23 | 19 |
| $N_{test}$ | 50 | 500 | 15 | 5 | 2 | 30 | 49 | 23 | 19 |
| Status | H | H | H | H | H | MS | T | H | AD |
| Age | $50 \pm 17$ | $29 \pm 4$ | $31 \pm 14$ | $31 \pm 7$ | $34 \pm 5$ | $43 \pm 12$ | $60 \pm 9$ | $76 \pm 5$ | $75 \pm 8$ |

a deformable registration step to normalize the brain shape. To do so, we extract the brain mask of all subjects as well as of the reference template, and realign each subject's brain mask to the reference template brain mask by solving the registration problem of Equation 1. The computed deformable transforms are then applied to the corresponding images to create shape-normalized brain datasets that can be used to train any network.

## 3. Experiments

### 3.1. Data

Seven T1w MRI datasets are used in our experiments: the five public healthy databases IXI[1], HCP[2] (Babayan et al., 2019), kirby (Landman et al., 2011), MPI (Babayan et al., 2019) and IBC (Pinho et al., 2018), the OFSEP/EDMUS multiple sclerosis (MS) dataset[3] from the "Observatoire français de la sclérose en plaques", the MS french registry (Vukusic et al., 2020; Confavreux et al., 1992), the MICCAI BraTS 2020 glial tumors public dataset (Bakas et al., 2017, 2018; Menze et al., 2014) that also includes the manual tumors segmentation, and the Alzheimer's disease (AD) ADNI-1 dataset[4] (Weiner et al., 2010) which also includes healthy subjects (CN). Division in training, validation and test sets is given in Table 1.

### 3.2. Experimental protocol

MR images are preprocessed using FSL FLIRT affine registration on the T1 MNI atlas (Jenkinson and Smith, 2001; Jenkinson et al., 2002), HD-BET brain extraction (Isensee et al., 2019) and N4 bias field correction (Tustison et al., 2010) except for the BraTS dataset. As this dataset is provided preprocessed using the CaPTk pipeline[5] that includes the brain extraction (different from HD-BET), we only applied the affine registration and the bias field correction. The final image size is $91 \times 109 \times 91$ with a 2mm voxel size. Binary classifiers are trained to classify the brain MRI datasets (either healthy, with multiple sclerosis or with tumor subjects). In the following, "shape normalized" datasets refer to the datasets normalized using the procedure of Section 2.3. We also evaluated the impact of elastic deformations data augmentation during training (denoted as "Elastic DA"). The deformations were chosen to be strong enough to hide the differences between brain shapes of the different datasets. Classification performances are evaluated using the true positive/negative rate (TPR/TNR) and the balanced accuracy (BA). In Section 4.2, we analyze

---

[1] brain-development.org/ixi-dataset    [2] humanconnectome.org/study/hcp-young-adult    [3] ofsep.org
[4] adni.loni.usc.edu    [5] cbica.github.io/CaPTk/preprocessing_brats.html

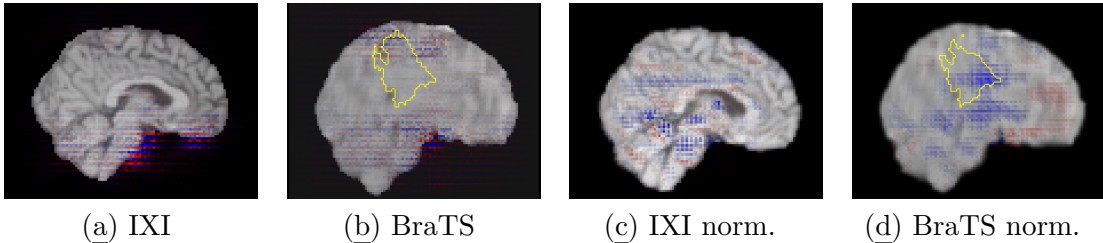

| (a) IXI | (b) BraTS | (c) IXI norm. | (d) BraTS norm. |

Figure 1: IXI vs BraTS classification gradient attributions. From left to right: for an IXI image, for a BraTS image, for a shape normalized IXI image and BraTS image. The tumor is in yellow, negative attributions in blue and positive ones in red.

the feasibility of distinguishing the datasets using only the confounding factor that-is-to-say using only the brain binary masks. In Section 4.3, we evaluate if the brain shape is indeed a confounding factor for classification models trained with MRI input[6].

### 3.3. Implementation details

The classifier, implemented in Pytorch, is a 3D PatchGan (Isola et al., 2017), trained with the Adadelta optimizer (Zeiler, 2012), class balanced minibatches, an initial learning rate set to 1 and the cross entropy loss. This CNN is defined as $C64$-$C128$-$C256$-$C512$ where $Ck$ denotes a Convolution-BatchNorm-LeakyReLU (slope 0.2) layer with $k$ filters, except for the first layer on which no BatchNorm is applied. At the end, a convolution is applied to obtain a 1-dimensional output. In Section 4.1, attributions were computed using gradient maps (Simonyan et al., 2013). The brain shape registration algorithm of Section 2.2 is based on the algorithm described in (Sdika, 2008). We set $t = 0.85$, $\lambda = 10^{-3}$ and the transformation is represented by a B-Spline vector field with a node spacing of 4 voxels.

## 4. Results

### 4.1. Attributions highlight brain borders

To visualize the confounding factors, we use attribution maps which indicate the relevance of each voxel in the network decision. Figure 1 shows some attribution examples for the IXI vs BraTS classification. In Figures 1(a) and 1(b), attributions are focused at the top and bottom of the brain, near the borders. Especially, high attributions (in absolute value) are in the brain stem. Yet, we might expect the areas inside the brain to be the most useful for decision-making. Although these attributions indicate that the brain shape can be involved in the network decision, they are not sufficient alone to draw a definitive conclusion.

### 4.2. Brain masks can be used to classify datasets

We apply the first step of our method as described in Section 2.2: binary classifiers are trained to classify several datasets using the brain masks only as input. As shown in Table 2, the first startling result is that, except for the intra-dataset ADNI task, it is possible to distinguish all pairs of datasets based on the brain shape only, without using any tissue or

---

[6] Healthy vs healthy datasets results are in the supplementary material for space consideration

Table 2: Classification accuracy on brain masks. The left dataset is the negative class.

| Classification task | | | TNR | TPR | BA |
|---|---|---|---|---|---|
| IXI | vs | BraTS | 1.00 | 1.00 | 1.00 |
| IXI | vs | OFSEP | 0.54 | 0.70 | 0.62 |
| IXI | vs | HCP | 1.00 | 1.00 | 1.00 |
| IXI | vs | IBC/kirby/MPI | 1.00 | 0.86 | 0.93 |
| HCP | vs | IBC/kirby/MPI | 0.99 | 0.95 | 0.97 |
| ADNI-CN | vs | ADNI-AD | 0.35 | 0.74 | 0.55 |

texture information. Brain masks from the tumors dataset BraTS can be classified from the healthy dataset IXI with a perfect classification score. Note that this classification performance could be partly explained by the difference between the BraTS preprocessing pipeline and the pipeline used for the other datasets. Classifying MS brain masks from the healthy ones is more difficult but it is still possible with a balanced accuracy of 62%. Thus, it is possible to classify healthy vs pathological subjects using only the brain shape. One might wonder to what extent the difference in brain shape is due to the pathology itself or to some differences due to the way datasets are built. To investigate the dataset construction effect, we consider the brain mask classification task between several healthy datasets and within the ADNI dataset. For example, we obtain a perfect accuracy for the IXI vs HCP problem. Even when several databases are aggregated in one class, the distinction is always possible with an accuracy higher than 85% for IXI vs IBC/kirby/MPI. Population age in the databases is another element that can influence brain shapes (with normal aging brain atrophy). However, despite the subjects of HCP and IBC/kirby/MPI match in age, it is possible to classify these datasets almost perfectly. In the intra-ADNI experiment, classes are from the same dataset and match in age but one is healthy and the other pathological. One can see that, despite the possible atrophy due to AD, brain masks are more difficult to distinguish with a balanced accuracy as low as 55%. This reinforces the idea that in general, the way datasets are built is a stronger factor than the disease itself for the brain mask classification. Thus, it is possible to distinguish various databases based on the brain shape only and the difference seems to be linked to a dataset construction effect that is not eliminated when disease or age factors are not present.

### 4.3. Brain shape is part of the decision

In this part, we applied the second step of the process described in Section 2.2: for each subject, a brain mask is randomly drawn from the opposite class and used to crop the current image ("Cropped images") or realigned its brain shape with a deformable registration ("Registered images"). An image for which the crop or the registration changes the predicted class, as well as the corresponding modified images, are shown in Figure 2. We can see that the brain shapes of the two classes are different. With the crop, the shape is partially modified: for example, the area around the stem or the frontal lobe do not change, whereas the registration allows to better fit the shape. Most of the tumor area is left untouched with the crop and it is probably not this loss of information that changes the classification. Indeed, on average in the test set (without shape normalization), only $3.6\% \pm 5.0\%$ of the tumor is cropped ($3.3\% \pm 5.5\%$ for the images still classified as patho-

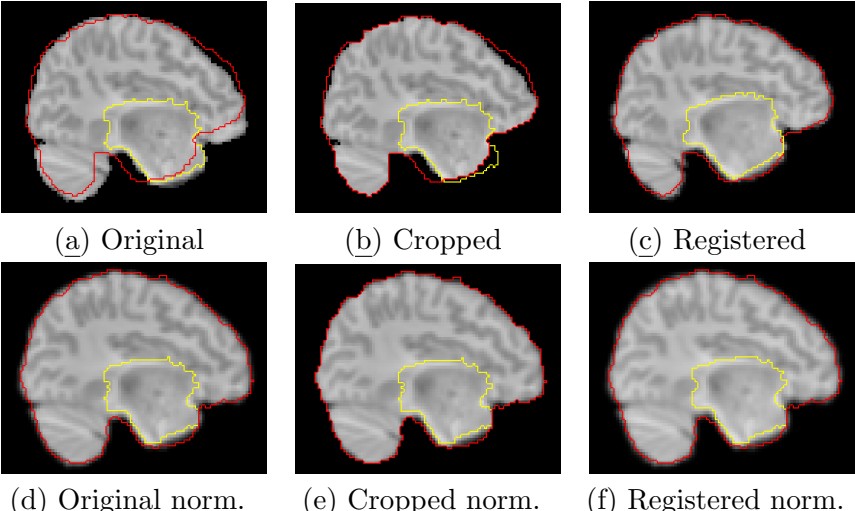

(a) Original     (b) Cropped     (c) Registered

(d) Original norm.     (e) Cropped norm.     (f) Registered norm.

Figure 2: Example of BraTS image (with tumor in yellow) with a random IXI brain mask (in red) and the corresponding modified images (cropped or registered). First (resp. second) line is without (resp. with) shape normalization. On this case, the original image without shape normalization is classified as pathological, modified images are classified as healthy.

logical and $5.6\% \pm 3.3\%$ for the others). The classification results are presented in Figure 3. For tumors, without elastic data augmentation, the classification performances on the original images are perfect. Conversely, when the shape is modified the accuracy of both classes falls. The TNR is lower than 50% for the cropped images and it falls to 16% for the registered images. Thus, the brain shape seems to be a key factor learned by the network to classify the images. For MS, the impact is not as strong but still present with a mean loss of accuracy of 3 points for cropped images and 16 points for registered images. This is consistent with the fact that brain masks are harder to distinguish for the MS dataset as shown in Section 4.2. When elastic data augmentation is used, the classification is more difficult on the original images for the tumors dataset but the decision seems less based on the brain shape as the accuracy decreases less when the modified images are tested: the accuracy is in average 27 points lower for cropped images and 7 points lower for registered images. This data augmentation improves the robustness, especially for MS as the classification is slightly better on the original images. In this case, the brain deformation seems to have hardly any impact with only 3-point accuracy difference on the registered images.

### 4.4. Deformable registration removes the confounding factor

The previous experiments validate that the brain shape is a part of the decision for MRI classification. We advocate that if the images are realigned not only with an affine registration but also such that the brain shapes are realigned to the reference template, the network decision can be free from the brain shape confounder. Figures 2 and 3 present the results of the same experiments as in Section 4.3 but using images with the shape normalization of Section 2.3. Visually, with the shape normalization, there is virtually no longer difference

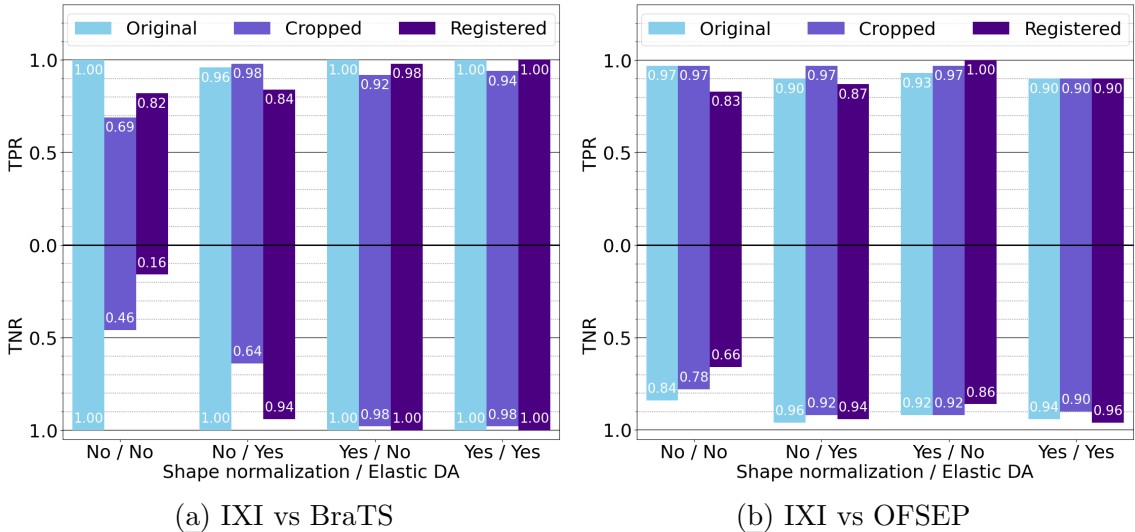

(a) IXI vs BraTS  (b) IXI vs OFSEP

Figure 3: True positive/negative rate (TPR/TNR) for IXI vs BraTS (left) or OFSEP (right) classification. Bar plots are grouped depending on whether brain shape normalization and elastic data augmentation are used or not. Colors refer to whether original images (light blue), cropped images (middle blue) or registered images (dark blue) are used at test time.

between the shapes. In terms of classification, we obtain similar or better performances than without normalization for both tumors and MS. When the modified images are used at the inference, the impact is minor, with an accuracy loss of around 5 points for tumors cropped images and around 1 point for tumors registered images. This is inferior to the model using images without shape normalization, even with elastic data augmentation. For MS, the results are equivalent or slightly better than without shape normalization and with elastic data augmentation. With the shape normalization, elastic data augmentation seems less useful. Moreover, in Figures 1(c) and 1(d), attributions are no longer localized on the borders but all over the brain. Therefore, the shape normalization seems to be enough to eliminate the confounding effect of the brain shape.

## 5. Conclusion

In this work, we propose a generic method to assess whether a variable is a confounding factor or not. We apply the proposed protocol to several public MRI datasets to identify the brain shape as a non-intuitive confounder for brain scans classification. In addition, we proposed to add a non-rigid brain shape realignment in the preprocessing pipeline to eliminate the confounding effect of the brain shape. As this step does not degrade the classification performances, our recommendation is to systematically use it (even when the brain shape is not a confounder) in addition to the affine registration conventionally used in standard pipelines. The elements highlighted in this paper could also be used in state-of-the-art methods like (Zhao et al., 2020; Wang et al., 2018), which so far have only been applied to solve the problem for scalar confounding variables. For this, the confounding variable predicted in these methods would be the brain mask through a segmentation loss.

## Acknowledgments

This work was supported by the LABEX PRIMES (ANR-11-LABX-0063) of Université de Lyon, within the program "Investissements d'Avenir" operated by the French National Research Agency (ANR) and by the "Projet Emergence" APIDIFF, CNRS-INS2I. We acknowledge the "Observatoire Français de la Sclérose en plaques" (OFSEP) for providing the data collected with ANR-10-COHO-002. This work was performed using HPC resources from GENCI-IDRIS (AD011012544/AD011012589).

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

## Appendix A. Visual results on MS

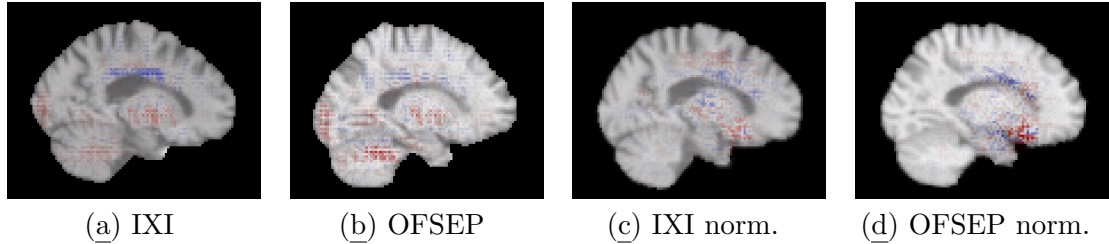

(a) IXI  (b) OFSEP  (c) IXI norm.  (d) OFSEP norm.

Figure 4: IXI vs OFSEP classification gradient attributions. From left to right: for an IXI image, for an OFSEP image, for a shape normalized IXI image and OFSEP image. Negative attributions are in blue and positive ones in red.

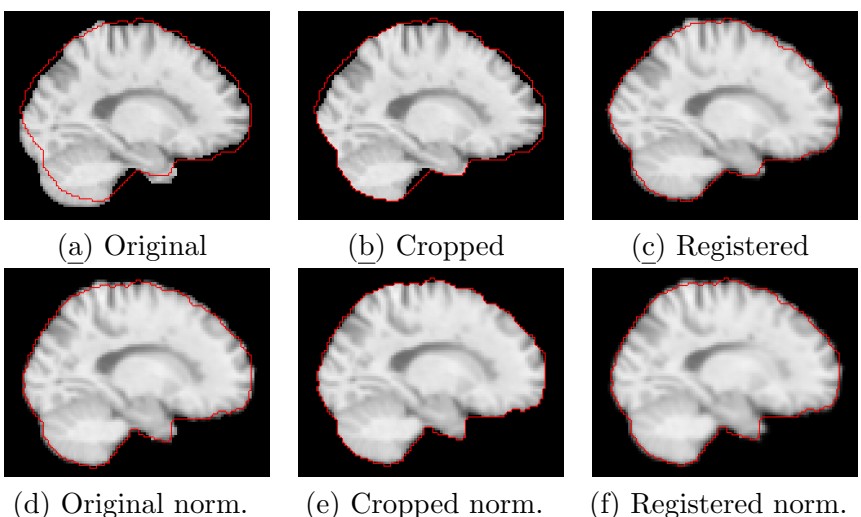

(a) Original  (b) Cropped  (c) Registered

(d) Original norm.  (e) Cropped norm.  (f) Registered norm.

Figure 5: Example of OFSEP image with a random IXI brain mask (in red) and the corresponding modified images (cropped or registered). First (resp. second) line is without (resp. with) shape normalization. In this case, the original and cropped images without shape normalization are classified as pathological, registered images are classified as healthy.

In Figures 4 and 5, we display attributions maps and shape modifications as in Figures 1 and 2, but for multiple sclerosis. The brain shape influence is less visible on attributions for multiple sclerosis than for tumors which is in accordance with numerical results of Sections 4.2 and 4.3. Indeed, high attributions are localized all over the brain. Using the brain shape normalized dataset changes the attributions localization: the decision seems less focused on the occipital lobe and the cerebellum, and more around the ventricles. The brain shape difference between the two datasets appears located at the back of the skull which is in line

with the attributions without shape normalization. The shape normalization is as efficient than on the tumors dataset, as the shapes of the two databases match better.

## Appendix B. Quantitative results on brain shape normalization

Table 3: Average brain volume (in voxels) with and without shape normalization.

| Dataset | Without normalization | With normalization |
|---------|----------------------|--------------------|
| IXI | 226695 ± 15393 | 233847 ± 3816 |
| OFSEP | 223919 ± 15816 | 233553 ± 4745 |
| BraTS | 232094 ± 5792 | 234053 ± 527 |
| HCP | 226818 ± 14623 | 234016 ± 4116 |
| IBC | 234565 ± 11071 | 235009 ± 624 |
| kirby | 223192 ± 10383 | 234477 ± 483 |
| MPI | 219067 ± 3165 | 234044 ± 470 |

In Table 3, we compared the mean brain volume between the datasets with and without the shape normalization proposed in Section 2.3. We can see that there is much less variability between the datasets and within the same database with the normalization, as desired.

## Appendix C. Second step on the healthy databases classification

In Figure 6, the classification results on the grayscale images for several healthy vs healthy databases classification are presented. The results show that, for the IXI vs HCP and the HCP vs IBC/kirby/MPI classifications, the brain is a confounder used by the network to make its decision. Indeed, when a shape transform is applied at test time, the classification performances fall. For the IXI vs HCP classification, the brain shape is no longer a confounder with either elastic data augmentation or the proposed shape normalization. For the HCP vs IBC/kirby/MPI classification, our brain shape normalization eliminates the brain shape confounder more efficiently than the data augmentation with a 5-point accuracy gain for the Cropped images. For the IXI vs IBC/kirby/MPI classification, the brain shape does not seem to be part of the network decision as the accuracy falls only slightly when the shape transformations are applied. Note however that even then, the shape normalization removes this slight accuracy decrease. As even when the brain shape is not (or barely) used as a confounder, the normalization does not degrade the performances and as the brain shape could be used to distinguish the databases (as shown with the first step in Section 4.2) in a different setup, the brain shape normalization seems to be a be a good step to add in a preprocessing pipeline.

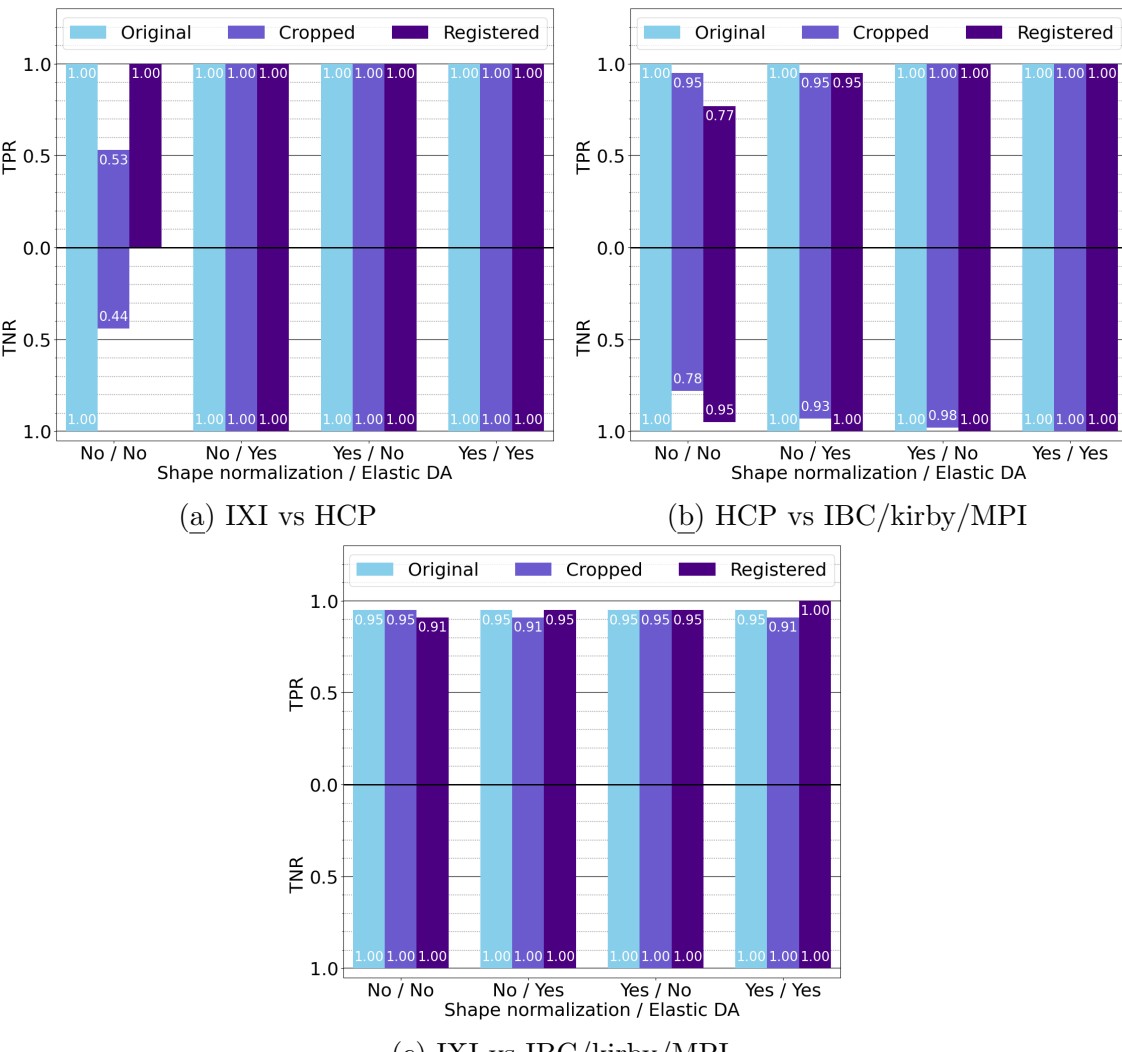

(a) IXI vs HCP     (b) HCP vs IBC/kirby/MPI

(c) IXI vs IBC/kirby/MPI

Figure 6: True positive/negative rate (TPR/TNR) for three healthy vs healthy classification. Bar plots are grouped depending on whether brain shape normalization and elastic data augmentation are used or not. Colors refer to whether original images (light blue), cropped images (middle blue) or registered images (dark blue) are used at test time.

