# OpenReview forum: "An unexpected confounder: how brain shape can be used to classify MRI scans ?"
_MIDL.io/2024/Conference — MIDL 2024 Oral_

### Official Review · Reviewer_YD1y · 2024-02-26

**Confidence:** 5
**Preliminary Rating:** 4
**Recommendation:** Poster
**Final Rating:** 4

**Summary:**

The paper proposes several methods to assess the importance of the brain shape (bias) in classification tasks based on T1-weighted MRI datasets and a solution to get rid of this bias. Large databases from healthy, MS and tumor’s datasets are used to train binary. After several tests (classification on original images, cropped images and images transformed with deformable transformation; generation of saliency maps), the authors conclude that brain shape is indeed used by the classifier to take the decision while it does not contain any medically relevant information about the pathology. Finally, the authors show that performing non-linear registration as a preprocessing step allows to obtain less biased classifiers.

**Strengths:**

The paper is well written and easy to read and follow. Large and diverse databases are used. The methods are described in details and appropriate. The conclusions are strongly supported by the results.

**Weaknesses:**

Although the methods for identifying confounders and correcting for it (for brain shape specifically) are proposed, no discussion is made about the importance and impact of the architecture of the classification model chosen (see Questions To Address In The Rebuttal).

**Detailed Comments:**

Introduction (end of 1st paragraph): “for pure pure classification problems” -> remove pure
Data: “Seven T1 MRI” -> should be T1-weighted

**Justification Of Final Rating:**

Thank you for the clarifications and updates. I think that the paper brings valuable insights to the community about common biases present in our datasets and will lead to interesting discussions around that topic at MIDL.

**Justification Of The Preliminary Rating:**

The paper is well written and the information it conveys is very important for the community to be aware of potential existing biases and to account for them in future models. The methods are detailed enough for others to easily reproduce the results and proposed solutions for bias mitigation.

**Questions To Address In The Rebuttal:**

The authors decided to use a 3D PatchGan model as classifier. It would be good to add a justification for the choice of that model, which does not seem to be very commonly used for brain MRI classification tasks.

Is it possible that the choice of model also influence the strength of the bias? Have you tried different model architectures and made the same observations?

Although affine registration to an atlas was performed on all databases, substantial differences in brain volume remain between databases that were processed with the authors pipeline (Tables 3, IXI and OFSEP). Did the authors do any evaluation of the registration quality?

**Special Issue:**

No

---

> ### Author Response · Authors · 2024-03-16
>
> **General comment:**
>
> We would like to thank all the reviewers for their comments, which helped improve the document.
> The manuscript modifications appear in color in the new submission.
>
> We corrected some misprints: the number of test samples is 500 and not 600 for HCP. The TNR and TPR in Table 2 for the HCP vs IBC/kirby/MPI classification were incorrect: the correct ones are TNR=0.99 and TPR=0.95 (instead of TNR=1.00 and TPR=0.82).
> The conclusion does not change and it is even strengthened since we can almost perfectly distinguish healthy datasets with matching age.
>
> As suggested by the reviewers, step 1 experiment for the ADNI dataset was added in the manuscript and step 2 experiments for the healthy datasets were added as supplemental material.
>
> **Response to the reviewer:**
>
> * On the choice of PatchGAN
>
> We tried to use other classical architectures (ResNet, DenseNet, MobileNet, etc) but we had overfitting problems on some of our small datasets. PatchGAN (a simple 4-layer CNN) was robust on all dataset classifications. The choice of the architecture can indeed have an impact on the confounder strength in the network decision but like the other hyperparameters (regularization, preprocessing, etc). The influence of the architecture would be interesting to investigate in an extended version, with the difficulty of finding a trainable architecture on our small datasets. However, our paper highlights the brain shape as a possible confounder for brain scans classification. The proposed methodology to detect and remove the brain shape confounder would be applicable to any architecture.
>
> * On the registration quality
>
> The affine registration has only been visually inspected on our databases. We indeed rely on the robustness of the chosen software (FLIRT) which is an established registration procedure in the community. As the affine registration only performs scaling and rotation, the brain shapes cannot perfectly match and it seems normal to have these volume variations.
>
> * Typos
>
> We corrected the identified typos.

---

### Official Review · Reviewer_P2iE · 2024-02-28

**Confidence:** 4
**Preliminary Rating:** 2
**Final Rating:** 4

**Summary:**

The paper investigates into the issue of confounding factors in models used for the classification of brain MRI scans, with a focus on the previously unconsidered factor of brain shape. It introduces a two-step method to first identify and then mitigate the confounding influence of brain shape on classification accuracy through the use of deformable registration. The approach aims to enhance the reliability of MRI classification outcomes by ensuring that decisions are based on clinically relevant features rather than spurious confounders. The authors test this and compare across different datasets and populations.

**Strengths:**

The paper introduces a new perspective by identifying brain shape as a potential confounding factor in MRI tasks, a variable not previously explored in depth.

The two-step protocol for confounder identification is interesting, providing a clear framework for assessing and addressing confounders in medical image analysis.

Proposing deformable registration as a preprocessing step to normalize brain shape is a practical and potentially effective solution to mitigate potential confounding effect, suggesting enhanced future models' clinical performance.

**Weaknesses:**

Some conclusions in the paper are reached without sufficient evidence. For example, the authors pose the question "whether the difference in brain shape we observe is due to the pathology, and therefore clinically relevant, or due to some differences in the population used to build the datasets." They then find that there exists a population difference and state "the difference seems to be linked to a population effect and not to relevant factors like disease or age." The fact that an observation is different between two populations does not shed any evidence at all on whether there is also a clinically relevant factor. The authors know this since the result of another experiment was that "it is possible to classify healthy vs pathological subjects using only the brain shape." Suggesting an association between the two.

Furthermore, it is well known that brain shape changes with age, and the authors mention this, contradicting the conclusion. I suggest the authors 1) rephrase this conclusion to the posed question and 2) clear up the seemingly contradictory phrasing, perhaps they are meant to be general vs specific but that is not very clear.

In terms of the proposed method to assess whether variables are confounders, it would be much stronger to show this method demonstrated on known confounders/non-confounders to show the efficacy of the method, before testing an unknown variable with a new method.

These weaknesses are fixable and I believe that there could be an interesting finding here but the current experimental set-up does not back-up the claims.

**Detailed Comments:**

The paper is well-written and structured. No minor comments, see weaknesses and question for more fundamental comments and issues.

**Justification Of Final Rating:**

According to the update from the authors, the paper now clearly describes the issues related to brain shape associations. Although another significant improvement would be testing on known confounders, I still believe it has its merits without those experiments and can lead to future research.

**Justification Of The Preliminary Rating:**

While the identified weaknesses in the paper appear to be amendable and the research could potentially yield intriguing results, the present experimental setup fails to sufficiently support the claims made. I look forward to the response and rebuttal by the authors.

**Questions To Address In The Rebuttal:**

Do the authors expect a different outcome had the images been aligned with the nonlinear MNI152 mapping vs the affine-only MNI used here?

There has been significant prior work on brain shape in the medical field. Pathology such as inflammation can have a direct impact on the shape of the brain [1], the effects of hypertension [2] or dementia [3]. How do the findings from this work align with the current literature on brain shape and pathology?

[1] O. Guillemot-Legris and G. G. Muccioli, “Obesity-induced neuroinflammation: beyond the hypothalamus,” Trends Neurosci, vol. 40, no. 4, pp. 237–253, 2017.

[2] Strassburger, Terri L., et al. "Interactive effects of age and hypertension on volumes of brain structures." Stroke 28.7 (1997): 1410-1417.

[3] Wang, Lei, et al. "Changes in hippocampal volume and shape across time distinguish dementia of the Alzheimer type from healthy aging." Neuroimage 20.2 (2003): 667-682.

---

> ### Author Response · Authors · 2024-03-16
>
> **General comment:**
>
> We would like to thank all the reviewers for their comments, which helped improve the document.
> The manuscript modifications appear in color in the new submission.
>
> We corrected some misprints: the number of test samples is 500 and not 600 for HCP. The TNR and TPR in Table 2 for the HCP vs IBC/kirby/MPI classification were incorrect: the correct ones are TNR=0.99 and TPR=0.95 (instead of TNR=1.00 and TPR=0.82).
> The conclusion does not change and it is even strengthened since we can almost perfectly distinguish healthy datasets with matching age.
>
> As suggested by the reviewers, step 1 experiment for the ADNI dataset was added in the manuscript and step 2 experiments for the healthy datasets were added as supplemental material.
>
> **Response to the reviewer:**
>
> * On the origin of the brain shape difference
>
> The conclusion was indeed not clearly stated in the initial submission and has hopefully been clarified. We did not (mean to) claim that age or disease are irrelevant for the brain mask classification, but that these two factors *alone* cannot explain the brain shape difference, and that there are multiple factors that we regroup under the term "dataset construction effect" (previously "population effect"). As we mentioned in the original manuscript, the brain shape difference can indeed be linked to multiple factors: disease (as in the papers the reviewer cites), age, sex, preprocessing pipeline, etc. To investigate the brain shape effect in the case of the classification of the healthy vs pathological images (MS or tumors), we considered the same classifier trained to distinguish healthy databases with similar age distribution. In this case, age or the considered pathology are not involved in the brain shape difference but the datasets can still be distinguished one from another. This implies that there are other factors, hard to identify and a priori unwanted for the healthy vs pathological classification, that impact the brain shape. Our intuition is that when matching datasets for factors such as age, sex or ethnicity, other unknown or unintuitive factors influencing the brain shape might remain, and the brain shape may still be a confounder.
> We reformulated the text for more clarity.
>
> * On applying our method on known confounders
>
> Thank you for the comments. It could be interesting to evaluate our method on a known confounder as the factors considered in [1]. It could be done in an extended journal version of this work.
>
> [1] *Sun, S., Koch, L.M., Baumgartner, C.F. (2023). Right for the Wrong Reason: Can Interpretable ML Techniques Detect Spurious Correlations?. In: Greenspan, H., et al. Medical Image Computing and Computer Assisted Intervention – MICCAI 2023. MICCAI 2023. Lecture Notes in Computer Science, vol 14221. Springer, Cham. https://doi.org/10.1007/978-3-031-43895-0_40*
>
> * On the MNI atlas
>
> Both the affine registration and the nonlinear brain shape normalization were done using the non-linear MNI atlas (MNI152_T1_2mm and not MNI152lin_T1_2mm). It seems that MNI152lin_T1 was removed from recent versions of FSL. We did not do the experiments to quantify the influence of the chosen reference template but our intuition is that this influence is negligible. There is a priori no reason for the spatial normalization (either the affine registration or the nonrigid registration on the brain shape) to be better or worse by choosing between two carefully built reference templates and that would influence the classification output.

---

> > ### Comment · Reviewer_P2iE · 2024-03-26
> >
> > I thank the authors for their detailed responses and clarification. I am updating my score to reflect that.

---

### Official Review · Reviewer_dJwQ · 2024-02-28

**Confidence:** 4
**Preliminary Rating:** 3
**Recommendation:** Oral
**Final Rating:** 4

**Summary:**

This paper discuss the potential for brain shape to be a confounder despite the standard affine spatial normalization, and try to prove that it is by attempting the possibilities to classify brain MRI scans into the database sources. In doing so, the authors propose two step methods:

(1) database source classification firstly using only the assessed confounder (brain shape) represented by 3D binary brain masks from different databases

(2) checking whether the brain shape is indeed a confounder by conventionally train the classifiers using the original brain MRI scans, then test the classifiers with two different transformation methods applied to the test data of the opposite class: (a) with and (b) without additional deformable registration. If the performance of (a) is lower than (b), it indicates that brain shape is indeed a confounder.

**Strengths:**

I like the idea and the clarity of this paper. This is a very important finding to recognize that brain shape can be a confounding factor. The paper also suggests an additional deformable registration more than just affine registration to address brain shape confounding, particularly when dealing with data from different databases. This can set a new standard for the community. I also appreciate that the authors try to validate the theory very carefully. The manuscript is well-written.

**Weaknesses:**

Although I found the overall methodology to be valid, I have concerns regarding the methodology’s first step to assess brain shape as a confounder. 3D binary brain masks are used to classify brain MRI database source in this step. The results show that these masks alone can distinguish database sources, even among datasets with similar demographics in terms of age and brain health condition. This raises important questions, particularly about the causes of such differentiation. My concern centers on the potential oversight of other indirect confounders (i.e. sex and/or race distribution within database) that could inadvertently influence the classification’s success.

At the same time, based on the findings of the methodology’s first step, the authors claim that age and brain health condition are irrelevant to the classification performance. This seems counterintuitive to the conventional understandings. Other than race and sex, age and brain condition are intricately linked to brain morphology and may act as indirect confounders, impacting the outcomes and interpretations of the study. It worries me if the authors come to the wrong conclusions for the first step’s results.

These concerns, however, do not diminish the validity of the results and conclusions derived from the methodology’s second step, which recognizes brain shape as a confounder. The suggestions following these concerns (see detailed comments) may help clarify the indirect effects of population demographics on brain shape, improving the study's conclusions.

**Detailed Comments:**

- Related to my comments on the weaknesses of the study, I would like to underscore several pivotal claims made in Section 4.2:
    - “Thus, it is possible to classify healthy vs pathological subjects using only the brain shape”
    - “…the difference seems to be linked to a population effect and not to relevant factors like disease or age.”

    These statements require further in-depth analysis to provide solid evidence. To enhance the credibility of these claims, I recommend the following considerations:

    - It is important to be transparent about disclosing potential confounding factors that could indirectly impact brain shape. This includes:
        - Providing details on the sex distribution within each database
        - Outlining the racial composition of subjects in each database.

        Such transparency is essential for a thorough evaluation of how these variables might influence the study's findings. In the absence of information on the sex and race of subjects, it is reasonable to assume that these factors may indeed influence the classification results, as you have stated. However, my primary concerns lie with the second claim regarding the insignificance of age and brain condition.

    - Conduct additional/supplementary analyses to solidify both claims (adding them to the methodology’s first step should be enough). I recommend intra-database comparison. Comparing brain conditions using only the brain masks within the same database (i.e. LGG vs HGG from BraTS). Although this should have been enough, comparing healthy vs pathological brain masks within the same database (e.g., CN vs AD from the ADNI dataset) may offer stronger evidence.
    - Reevaluate conclusions of the findings. I am concerned that the authors arrived at inaccurate conclusions, particularly for the methodology’s first step. Based on the results of aforementioned additional investigations, I recommend reevaluating the statements, especially the second claim, which contradicts traditional notions[1]. The first and second claims themselves appear to be inconsistent. It may be more accurate to acknowledge that factors such as age and brain condition may interact intricately, influencing brain morphology alongside race and sex. Acknowledging this does not reject the finding (particularly from the methodology’s second step) that brain shape acts as a confounder. Instead, it enriches our understanding of how other confounding factors may influence brain shape, and the paper gives solution to this matter. Maybe, this will also explains why classifying MS data from OFSEP vs normal data from IXI datasets is harder, even when compared to IXI datasets vs other normal datasets.
- While I find the methodology's second step valid, I noticed that in the experiments performed in sections 4.3 and 4.4, there is no mention of comparisons between normal datasets. Including such comparisons, particularly using the original 3D brain MR images between datasets like IXI vs. IBC or especially between HCP vs. IBC (which share similar age distributions), could enrich the study. It would offer additional insights into the influence of brain shape across different healthy populations, further contextualizing the role of brain shape as a confounder. Could the authors clarify their reasons for this exclusion?
- Minor comments:
    - In section 2.1, paragraph 1, the paper mentions: “If the classifier is random, with an accuracy of 0.5”. Does this mean that the accuracy has to be precisely 0.5 or do you mean lower than 0.5?
    - The caption in figure 2, “First (resp. second) line is without (resp. with) shape normalization.” Do you mean first line is without shape normalization, and second line is with shape normalization?


Reference:

[1] Zhao L, Matloff W, Ning K, Kim H, Dinov ID, Toga AW. Age-Related Differences in Brain Morphology and the Modifiers in Middle-Aged and Older Adults. Cereb Cortex. 2019 Sep 13;29(10):4169-4193. doi: 10.1093/cercor/bhy300. PMID: 30535294; PMCID: PMC6931275.

**Justification Of Final Rating:**

The paper has addressed the majority of the review comments and revisions have significantly improved the clarity and validity of the research. Although questions about what cause the brain shape confounders remain, this could spark more research in the future. Overall, the paper adds valuable insights to the field and deserves to be included in the conference.

**Justification Of The Preliminary Rating:**

Although the manuscript presents a valid and interesting approach to assessing the role of brain shape as a potential confounder in brain MRI classification and propose solution to this, this rating “3: Borderline” reflects significant concerns, particularly the validity of the findings from the methodology’s first step. The claim that age and brain health condition are irrelevant to the classification performance contradicts conventional neuroscientific understanding. Additionally, potential confounding factors such as racial and sex-related biases are not adequately considered, which could impact the study’s conclusions.

The research question, however, is of clear importance, and the overall methodology has a strong foundation. I believe that with additional analyses, a more thorough examination of brain shape relation with other potential confounders, and reevaluation of the study’s conclusions/claims, the manuscript could make a significant contribution to the field.

**Questions To Address In The Rebuttal:**

In the detailed comments, I highlighted the need for several considerations, which include:

- transparency in disclosing information of other potential confounding factors that might influence brain shape
- additional comparison and analyses to support the study’s conclusions and statement, especially related to the methodology’s first step
- reevaluation of the study’s conclusions due to potential oversight of other confounding factors that might have had influence the brain shape
- a rationale behind the exclusion of comparison between normal datasets in the methodology’s second step experiments

**Special Issue:**

Yes

---

> ### Author Response · Authors · 2024-03-16
>
> **General comment:**
>
> We would like to thank all the reviewers for their comments, which helped improve the document.
> The manuscript modifications appear in color in the new submission.
>
> We corrected some misprints: the number of test samples is 500 and not 600 for HCP. The TNR and TPR in Table 2 for the HCP vs IBC/kirby/MPI classification were incorrect: the correct ones are TNR=0.99 and TPR=0.95 (instead of TNR=1.00 and TPR=0.82).
> The conclusion does not change and it is even strengthened since we can almost perfectly distinguish healthy datasets with matching age.
>
> As suggested by the reviewers, step 1 experiment for the ADNI dataset was added in the manuscript and step 2 experiments for the healthy datasets were added as supplemental material.
>
> **Response to the reviewer:**
>
> * On the intra-database classification
>
> We have added an intra-database (ADNI CN vs ADNI AD) experiment for the first step. The results show that the classifier cannot distinguish these two sets based on the brain shape. Here, the disease does not seems to influence the brain shape. The Brats LGG vs HGG was not added or discussed for space reasons. In this case, the balanced accuracy is 57%.
>
> * About the claim that disease/age would be irrelevant for the classification
>
> The conclusion was not clearly stated in the initial submission and has hopefully been clarified. We did not (mean to) claim that age or disease are irrelevant for the brain mask classification. We indeed agree with the reviewer that disease (like MS) or age can modify the brain shape as mentioned in the original submission. Nevertheless, as it is possible to distinguish brain shape between healthy datasets with similar age distribution, it seems that these two parameters *alone* are unable to explain the difference of the brain shape between two databases (but are not irrelevant). The difference should come from a combination of factors that we regroup under the term of "dataset construction effect" (previously "population effect").  Our intuition is that when matching datasets for factors such as age, sex or ethnicity, other unknown or unintuitive factors influencing the brain shape might remain, and the brain shape may still be a confounder.
> We reformulated the text for more clarity.
>
> * On the sex/race influence
>
> Race was not disclosed in the datasets' information.
> The percentage of women, for each dataset (except for BraTS2020 as the information is not provided), is given below:
>
> IXI     : 65;
> HCP     : 59;
> MPI     : 19;
> kirby   : 45;
> IBC     : 17;
> OFSEP   : 68;
> ADNI-CN : 47;
> ADNI-AD : 48
>
> Age and disease were taken as examples for discussion on the factors that can influence the brain shape. Other factors can indeed be considered. Although a deeper discussion on other factors would be interesting, there is no space to add this discussion that could appear in an extended journal version of the work.
>
> * On the exclusion of comparison between normal datasets in second step experiments
>
> We did not do the step 2 experiments for the healthy vs healthy datasets for the initial submission. We chose to only do the second step for the healthy vs pathological (BraTS or OFSEP) classification because it seemed to us it has a more practical and clinical interest than classifying among healthy images. Nevertheless, to avoid the impression of "cherry-picking", we did the experiments since the reviewers feedback and added the results on the healthy databases classification in the Appendices of the new version.
>
> * On the random classifier
>
> We mean an accuracy close to 0.5 as this would mean the classifier is not able to distinguish from one class to the other.  A classifier with an accuracy much lower than 0.5 would mean that it can distinguish brain masks among different classes but would inverse the datasets: in this case, it is worth considering whether the training was conducted correctly.
>
> * On the Figure 2 caption
>
> Yes, the first line is without shape normalization and the second line is with shape normalization.

---

> > ### Comment · Reviewer_dJwQ · 2024-03-26
> > **Response to Rebuttal**
> >
> > Thank you for addressing the review comments and making revisions. I am updating my score to reflect the paper's merits for publication.
> >
> > One last suggestion/question: Based on the intra-database classification results from the ADNI dataset, it appears that brain masks alone cannot distinguish brain abnormalities within a single database. Does this indicate that additional deformable registration isn't needed for intra-database classification? Clarifying this could help readers understand when deformable registration is or isn't necessary, potentially simplifying future research methodologies

---

> > > ### Author Response · Authors · 2024-03-27
> > >
> > > The results seem indeed to indicate that the deformable registration is not necessary in this case. Nevertheless, the results in Figure 3 and 6 show that applying this spatial normalization to the images does not reduce the classification performances for all experiments. Therefore, as mentioned in Appendix C, our take-home recommendation is to apply it as a systematic preprocessing step. We added this recommendation in the conclusion.

---

### Official Review · Reviewer_ZnHV · 2024-02-29

**Confidence:** 4
**Preliminary Rating:** 4
**Recommendation:** Poster
**Final Rating:** 5

**Summary:**

The study investigates whether the brain shape is a potential confounder for brain MRI classification tasks. The authors use skull-stripped data from the IXI, HCP, MPI, kirby21, IBC, OFSEP, and BraTS2020 collections. In the first experiment, the brain masks are used to classify the origin dataset of scans. Later, cropping and registration to the target dataset brain mask are applied to demonstrate that the classification performance indeed drops at test time, indicating that the shape is relevant. Deformable registration and data augmentation with elastic deformation is shown to be able to partially mitigate the issue.

**Strengths:**

- Compelling new insight into potential new confounding factors for brain MRI classification
- Experimental setup is tailored to the research question at hand
- Experiments are documented such that reproducing the study should be straightforward
- Discussion about the impact and related de-confounding methods

**Weaknesses:**

- The method was tested on many different datasets, but the results are only presented/discussed for a small subset of the parts in 4.3/4.4. Please either add the full results in the supplementary or briefly discuss them to avoid the impression of "cherry-picking".
- The motivation for the choice of classifier is a bit unclear and unusual (PatchGAN). Since it is a patch-based approach, details regarding patch sizes, etc., are missing.

**Detailed Comments:**

I appreciate the clear experimental setup and the interesting research question you raise.

- Can you comment on how you split the dataset into train/val/test sets?
- The usage of a PatchGAN classifier seems a bit odd to me. Did you also consider other more commonly used networks for MRI classification (e.g., ResNets, DensNets)?
- Can you comment on the impact of the deformable registration on downstream tasks? For some applications, altering the brain shape may cover the underlying condition you are trying to classify.
- 2.1: "If the classifier is random, with an accuracy of 0.5, the variable can be discarded from the potential confounders of the problem."
-> only for a balanced dataset!
- I find your use of the designations "registered" and "shape normalized" for deformable registration a bit confusing, especially in the figures where you have registered and normalized cases.
- For the results in Fig. 3: If I read this correctly, which image type was used (original, cropped, registered) is indicated at test time. Re-iterating this here would add clarity.

Typos:
- Page 1: "confounding factor play**s** an important"
- Page 1: "pure ~pure~ classification problems"
- Opening quotation marks: You can do this in $\LaTeX$ with `` and '' to avoid getting closing marks at the beginning.

**Justification Of Final Rating:**

Thank you for your detailed replies to my comments and for your effort to improve the manuscript. The submission is now in my opinion significantly improved, important points clarified, and I therefore have adjusted my rating.

**Justification Of The Preliminary Rating:**

The study offers an interesting insight into brain shape as a confounding factor for brain MRI classification. The experimental setup is sufficiently described for the study to be reproducible. I appreciate that the authors tested it on many datasets; showing the results for all experiments (or briefly discussing them) would add confidence to the conclusions.

**Questions To Address In The Rebuttal:**

- Did you also test the grayscale-based confounding for the rest of the experiments listed in Table 2? With you only mentioning the IXI vs. OFSEP and IXI vs. BraTS this would be interesting
- Please provide more details regarding the selection of the PatchGAN classifier as mentioned above

**Special Issue:**

No

---

> ### Author Response · Authors · 2024-03-16
>
> **General comment**:
>
> We would like to thank all the reviewers for their comments, which helped improve the document.
> The manuscript modifications appear in color in the new submission.
>
> We corrected some misprints: the number of test samples is 500 and not 600 for HCP. The TNR and TPR in Table 2 for the HCP vs IBC/kirby/MPI classification were incorrect: the correct ones are TNR=0.99 and TPR=0.95 (instead of TNR=1.00 and TPR=0.82).
> The conclusion does not change and it is even strengthened since we can almost perfectly distinguish healthy datasets with matching age.
>
> As suggested by the reviewers, step 1 experiment for the ADNI dataset was added in the manuscript and step 2 experiments for the healthy datasets were added as supplemental material.
>
> **Response to the reviewer:**
>
> * On the choice of showing only the step 2 results for the healthy vs pathological classification
>
> We did not do the step 2 experiments for the healthy vs healthy datasets for the initial submission (it was not "cherry-picking").
> We chose to only do the second step for the healthy vs pathological (BraTS or OFSEP) classification because it seemed to us it has a more practical and clinical interest than classifying among healthy images. Nevertheless, to avoid the impression of "cherry-picking", we did the experiments since the reviewers' feedback and added the results on the healthy databases classification to the Appendices of the new version.
>
> * On the choice of PatchGAN
>
> We indeed tried to use other classical architectures (ResNet, DenseNet, MobileNet, etc) but we had overfitting problems on some of our small datasets. PatchGAN (a simple 4-layer CNN) was robust on all dataset classifications. The model is trained on the complete image. The "patch" name comes from the fact that the output is a reduced 3D maps of logits (patches are the fields of view of these logits). Each one of these logits is trained for the classification. At test time, we used the average of these logits to classify.
>
> * On the databases split
>
> We perform a random split based on 80/10/10% for the small databases and we adjust the number of samples for the bigger databases to avoid excessive imbalance at training time.
>
> * On the impact of the deformable registration
>
> In this work, we show that the brain shape can be used in a classification task. It can indeed be a clinically relevant factor or not. And even when it is a clinically relevant factor, we might also want to discard its influence to force the classifier to focus on other factors (lesions, internal structure, etc). Therefore, using the normalization or not is application depending, based on whether it is known that the brain shape is clinically relevant (or not). Note that, when the relevance of the brain shape is not known, our methodology can also be used to measure its influence in the classification. Note also that the proposed deformable registration is only performed to fit the borders and not the internal structures as we can expect from usual deformable registrations. These structures, their size and their shape are then mostly preserved by our registration.
>
> * On the random classifier
>
> During the training, the batches are balanced to cancel the effect of our imbalanced datasets (this point was added in the manuscript). At test time, when the classifier is random (return class 1 with a probability 0.5), both accuracy and balanced accuracy will be 0.5 whatever the balancing of the classes in the test dataset.
>
> * On the Figure 3 legend
>
> Indeed, the image type (original, cropped, registered) is at test time. We have added it to the legend.
>
> * Typos
>
> We corrected the identified typos.

---

### Comment · Area_Chair_jb99 · 2024-03-18
**Invitation to reply to authors**

Dear reviewers,

The authors have prepared responses to your comments, which you should now be able to see in OpenReview. We encourage you to reply to their comments, and where necessary, adjust your rating. Please do so before the 27th of March.

---

### Meta-Review · Area_Chair_jb99 · 2024-04-01

**Recommendation:** Accept (Oral)
**Confidence:** 4

**Metareview:**

The paper explores brain shape as a confounder in brain MR classification; this is in line with some recent attention to confounders/shortcuts in the medical imaging community. The reviewers agree on the relevance of the topic and clarity of the paper. During the discussion phase, the main concerns revolved around the validity of the reached conclusions based on the presented experiments and explanations. The authors have addressed this in the revision, with several reviewers raising their scores. I agree with several of the reviewers that this work would make an interesting contribution to the conference and am happy to recommend acceptance.

---

### Decision · Program_Chairs · 2024-04-06

Accept (Oral)